# OpenReview forum: "Feint and Attack: Attention-Based Strategies for Jailbreaking and Protecting LLMs"
_ICLR.cc/2025/Conference — ICLR 2025 Conference Withdrawn Submission_

### Official Review · Reviewer_nW6S · 2024-10-30

**Soundness:** 2
**Presentation:** 3
**Contribution:** 2
**Rating:** 5
**Confidence:** 4

**Summary:**

The paper explores the attention mechanisms of LLMs to enhance jailbreak attacks and defenses. Using three metrics for analyzing attention distributions (Attn_SensWords, Attn_DepScore, and Attn_Entropy), the paper develops a new attack (ABA) and a new defense (ABD), both based on the intuition that by diluting the attention weights on key tokens in the jailbreak prompts. The work demonstrates the important role played by attention mechanisms in jailbreak attacks/defenses.

**Strengths:**

- The work characterizes jailbreak attacks from the perspective of attention weights, which is overlooked in previous studies.
- The proposed attack outperforms a set of existing attacks in terms of ASR and number of queries.
- The paper is overall well-written and structured.

**Weaknesses:**

- The proposed metrics and methods lack any theoretical justification. Why are these particular metrics chosen? What are their connections/differences? While the proposed beam search algorithm for prompt refinement is interesting, it could benefit from a more detailed analysis of its convergence properties.
- The proposed defense (ABD) seems rather heuristic. Why is Attn SensWords not used in the formula? How to set the hyper-parameter $\sigma$ optimally? Further, can the adversary develop an adaptive attack to circumvent ABD?
- The experimental evaluation can be further improved. For example, using a larger dataset beyond 50 prompts, adding ablation studies to understand the impact of individual design choices (e.g., the interplay between different metrics), and evaluating false positive rates of the defense strategy.

**Questions:**

- Provide theoretical justification for the proposed metrics/methods.
- Conduct an empirical evaluation to understand the interplay between different metrics.

---

### Official Review · Reviewer_UvZG · 2024-11-03

**Soundness:** 1
**Presentation:** 1
**Contribution:** 2
**Rating:** 3
**Confidence:** 4

**Summary:**

This work proposes an attention-based attack and defense by observing that the attention weights can be correlated to the jailbreak ASR.

Specifically, it first defines three metrics to measure the attention weights: 1. AttnSenwords, which is the average attention weight given to sensitive words across all prompts, timesteps, layers, and attention heads, 2. AttnDepScore, which measures the average ratio of attention paid to input tokens versus all tokens (input + generated) during the text generation process, indicating how much the model relies on the input context when generating outputs, and 3. AttnEntropy, which calculates the average entropy of attention weight distributions across all timesteps, layers, and heads to measure how focused or dispersed the model's attention is across input tokens.

Then, the experiment of four jailbreak methods on two datasets, AdvBench and Dolly, shows ADS and AE are positively correlated to ASR, while ASW is negatively correlated to ASR. Thus, for Attention-Based Attack (ABA), the paper proposes to iteratively refines malicious prompts by minimizing attention weights on sensitive words and maximizing attention entropy through nested task generation, using beam search to select candidates with lowest attention scores until a successful jailbreak is achieved.

ABD defends against jailbreak attacks by detecting suspicious attention patterns through a weighted combination of dependency and entropy scores, then prepends safety assessment instructions to potentially harmful prompts.

Experiments on five models comparing to six baselines demonstrates the effectiveness of ABA. ABD is also illustrated to be effective in defending the baseline attacks as well as ABA.

**Strengths:**

1. This is the first work in analyzing jailbreak attacks through attention mechanisms, providing a fresh approach to understanding how these attacks work at a model behavior level.

2. Comprehensive attack success rates across multiple models, showing strong empirical results on both open and closed source models including the latest LLMs like GPT-4 and Claude-3.

**Weaknesses:**

1. The paper's claimed correlation between attention metrics and Attack Success Rate (ASR) is not well supported by the empirical results. For instance, in Llama2-7B, TAP and DeepInception have nearly identical Attn_SensWords values (0.0089 vs 0.0087) yet their ASR differs dramatically (0.30 vs 0.69), demonstrating that these metrics may not be reliable indicators of attack effectiveness. The authors should address such contradictory examples and provide more rigorous statistical analysis to support their claims about the relationship between attention patterns and attack success.

2. A key concern is that the metrics may be sensitive to differences in prompt structure introduced by different attack strategies (TAP, PAIR, etc.) rather than capturing meaningful patterns specific to jailbreak behavior. To address this, I suggest normalizing each metric by comparing values for sensitive words against non-sensitive words within the same prompt. Specifically:

Normalized ASW = (Attention on sensitive words) / (Attention on non-sensitive words)

Normalized ADS = (Dependency score for sensitive words) / (Dependency score for non-sensitive words)

Normalized AE = (Attention entropy over sensitive words) / (Attention entropy over non-sensitive words)

These normalized metrics would control for prompt-level variations introduced by different attack strategies and show whether sensitive words receive truly distinctive attention patterns.

3. A major limitation of the paper is the lack of concrete implementation details for the core prompt generation and refinement mechanisms. While the authors describe high-level concepts like "semantic-guided scenarios" and "nesting of multiple tasks," they fail to specify how these scenarios are actually generated, how tasks are nested, or what templates and patterns are used. The Refine() function in Algorithm 1 is particularly problematic as it provides no information about how the refinement process works or how it utilizes attention weights. Most critically, there are no examples of how an original malicious query is transformed through their refinement process, making it impossible for other researchers to reproduce or build upon this work.

      To improve the paper's contribution, I strongly recommend adding: (1) A detailed description of the prompt generation and refinement mechanisms, including specific rules or templates used; (2) Several concrete examples showing how an original query is transformed through multiple refinement iterations, with attention weights and entropy values at each step; (3) A clear specification of how new scenarios are generated while maintaining semantic relevance to the original query.

4. The adaptability of the parameter alpha to unseen jailbreak strategies is not addressed. The paper should discuss how this critical parameter would perform when encountering previously unseen attack patterns and whether it requires retraining.

5. The authors claim ABA's transferability to closed-source models through "effective scenario nesting templates" learned from open-source models. However, this significant claim lacks empirical validation. The paper should provide experimental results demonstrating this transfer learning capability, or remove the unsubstantiated claim.

6. The definition of the "Queries" metric lacks crucial clarity. While described as "the average number of successful jailbreak attacks between the attack model and the target model," this definition is ambiguous and seems inconsistent with the results. The data shows methods with vastly different Attack Success Rates (ASR) having similar Query counts (e.g., ReNeLLM with 71.8% ASR and ABA with 98.4% ASR have similar Queries of 3.9 and 3.6), which raises questions about what exactly is being averaged. The authors should explicitly specify: (1) whether Queries counts attempts until first success or all attempts including failures, (2) if there is a maximum attempt limit, and (3) how failed attempts are factored into the metric. Without this clarification, the Queries metric cannot be meaningfully used to compare the efficiency of different attack methods.

7. Lack of robustness test. How does ABA perform against jailbreak defense techniques, such as paraphrasing[1], SmoothLLM [2], Backtranslation[3].
[1] Jain, Neel, et al. "Baseline defenses for adversarial attacks against aligned language models." arXiv preprint arXiv:2309.00614 (2023).
[2] Robey, Alexander, et al. "Smoothllm: Defending large language models against jailbreaking attacks." arXiv preprint arXiv:2310.03684 (2023).
[3] Wang, Yihan, et al. "Defending llms against jailbreaking attacks via backtranslation." arXiv preprint arXiv:2402.16459 (2024).

And how does ABD perform comparing with them?

8. The paper could include sections discussing the societal impact and limitation.

Minor:
1. The presentation of tables would benefit from clearer formatting conventions. Column headers should include "(%)" where values are percentages, and table captions would be more informative if they specified the metrics being presented (e.g., "ASR results for different defense methods" rather than "The results of different defense methods").

2. The Related Work section would be more appropriately placed in the main text rather than the appendix, as it provides crucial context for understanding the paper's contributions and positioning within the field.

3. The overall presentation could be improved. For example, the description sometimes is confusing and vague. See questions below.

**Questions:**

1. Regarding attention weight calculation: for a sensitive word at position i, are you measuring:

a) The attention it receives from other tokens (when i is a key position)?

b) The attention it pays to other tokens (when i is a query position)?

c) Or some combination of both?

2. Regarding the timestep dimension T: What is the scope of timesteps in your calculation? Does T represent:

a) Only the positions in the input prompt?

b) Both input prompt and generated output positions?

c) Different ite rations/forward passes of the model?

   What is N in eq 2 and eq3? Is it intended to be M because M is defined as the length of input before but never used.

Is padding or truncation applied when calculating all three metrics?

2. In Fig.2, by “the x-axis represents the index of the sentences from the input of the model”, do you mean: 1. each point on the x-axis represents a sentence position within a single multi-sentence input (e.g., sentence 1, 2, 3 from the same prompt)
OR 2. each point represents different input samples/prompts from the dataset.

    This distinction is important for interpreting the visualization and should be clarified in the figure description. The current description "the index of the sentences from the input of the model" could be interpreted either way.

---

### Official Review · Reviewer_QTp2 · 2024-11-03

**Soundness:** 3
**Presentation:** 3
**Contribution:** 3
**Rating:** 5
**Confidence:** 5

**Summary:**

The paper introduces an attention-based framework for both attacking and defending large language models (LLMs) by manipulating attention distributions. The authors propose an innovative "Feint and Attack" strategy, the Attention-Based Attack (ABA), which diverts the model’s attention away from harmful content by embedding nested, benign prompts. This method is paired with the Attention-Based Defense (ABD), which uses attention metrics to detect and mitigate potential jailbreak attacks. The metrics introduced, such as Attn SensWords, Attn DepScore, and Attn Entropy, contribute to understanding how attention shifts influence model output, providing insight into the LLMs' vulnerability to adversarial manipulation.

**Strengths:**

Novel findings:
The finding of the attention mechanisms around jailbreak attacks is novel. And it offers a new perspective on how to optimize the jailbreak attacks and defenses in the future.
High effectiveness:
The Attention Based Attacks seems to outperform current baselines in jailbreak attacks in terms of attack success rate.
Thorough analysis:
The authors introduce three different metrics for analyzing the relationships between attention and jailbreak attacks. The new metrics offer new perspectives in understanding the attacks and potentially how to defend the attacks.

**Weaknesses:**

ABD weakness 1:
This jailbreak defense is built on an attention-based risk score. And as mentioned in the paper, a suitable threshold is the foundation of ABD.  In general, a good metric/threshold comprises two components: high True Positive Rate (TPR) and lower False Positive Rate (FPR). Table 3 shows that ABD can have pretty high TPR when testing on datasets containing evil targets only. I am curious to see if the TPR will drop or the FPR will rise when you mix in 50% benign prompts into testing. My concern is that if the risk threshold is set to be too rigorous, the warning will appear most of the time when you receive a prompt, regardless harmful or not. So it might better to show the trade-off between TPR and FPR of this defense, demonstrating with a ROC curve might be ideal.

ABD weakness 2:
Assuming that the risk score metric can perfectly detect all evil contents, adding warning prefixes such as “Attention! The following content …” is equivalent to existing defense methods such as self-reminder (https://www.nature.com/articles/s42256-023-00765-8)  or goal_priority (https://arxiv.org/abs/2311.09096) . In other words, why not add these warning prefixes to all prompts that come in? It is one of the most commonly used approaches and has been proven effective.

ABA weakness 1:
The ASRs of Baselines of the ABA seems a bit off:
TAP reported 4% ASR attacking Llama2-7B in its paper while in the same paper PAIR has 0% ASR attacking Llama2-7B, which makes sense because they were black-box attacks that were supposed to be weaker. However, in this paper they both have around 30% which I have never encountered in practice. On the contrary, AutoDAN normally has an around 60% ASR testing on the full Advbench while it only has less than 30% here in this paper. Such a huge difference makes me a little concerned about the subset of Advbench being used in this paper.
ABA weakness 2:
Although the paper emphasizes the three metrics Attn SensWords, Attn DepScore, and Attn Entropy, they are not used in the implementation of ABA. I am curious if the authors have explored the option of integrating these metrics as part of the loss function?

**Questions:**

check weakness

---

### Official Review · Reviewer_T7Lr · 2024-11-04

**Soundness:** 1
**Presentation:** 2
**Contribution:** 2
**Rating:** 3
**Confidence:** 4

**Summary:**

This paper explores jailbreak and defense strategies for LLMs using attention-based methods. Three statistical analysis approaches are proposed: Attention Intensity on Sensitive Words, Attention-based Contextual Dependency Score, and Attention Dispersion Entropy. These metrics serve as the foundation for designing effective jailbreak and defense strategies. The proposed attack and defense methods have achieved good results.

**Strengths:**

1. The topic is important for LLM security.
2. The attention-based analysis is interesting.

**Weaknesses:**

1. The preliminary analysis in Section 2 lacks sufficient breadth to fully support the authors' arguments and the attack and defense method design.
2. The compared jailbreak methods are all proposed in 2023 and may not reflect the latest developments in the field. There are many jailbreaking attacks proposed in 2024, and they work well.
3. Many descriptions lack detail, leading to potential confusion.
4. Lack of comparison with other defense methods.

**Questions:**

1. There have been many new jailbreak techniques introduced in 2024.  Why these methods have not been included in the analysis? Additionally, why were GCG and its optimized methods not included in Table 1 for comparison? Furthermore, Table 1 should not include an analysis of the proposed ABA attack method in this paper, as doing so leads to a result-oriented argument, which is a flawed logical approach.
2. Why AE and ASR are positively correlated, with higher AE leading to higher ASR? If the model is more persistent in responding to jailbreak questions, the AE should be lower, yet ASR could remain high. For instance, in methods like GCG that use loss-oriented optimization, is it possible to achieve high ASR with low AE? However, the authors did not show this in Table 1.
3. The ASR results for TAP and PAIR are surprisingly low. Could you confirm if the optimal settings for TAP and PAIR were used to generate the jailbreak prompts? Additionally, it would be helpful if the authors could provide some example jailbreak prompts from the different jailbreak methods.
4. How many prompts are used in Table 1?
5. What is the jailbreak method for Figure 2?
6. The Attention-based Contextual Dependency Score is calculated based on both the input and output. In the defense strategy described in Section 3.2, is a manually crafted jailbreak response used as the output to compute the Attn_DepScore? Maybe the target goal of AdvBench to calculate the Attn_DepScore?

---

### Note · Authors · 2024-11-13

I have read and agree with the venue's withdrawal policy on behalf of myself and my co-authors.